

# Maxwell-Stefan diffusion: a framework for predicting condensed phase diffusion and phase separation in atmospheric aerosol

Kathryn Fowler[1], Paul J. Connolly[1], David O. Topping[1], and Simon O'Meara[1]

[1]University of Manchester, School of Earth, Atmospheric and Environmental Science

*Correspondence to:* Paul James Connolly (paul.connolly@manchester.ac.uk)

**Abstract.**

The composition of atmospheric aerosol particles has been found to influence their micro-physical properties and their interaction with water vapour in the atmosphere. Core-shell models have been used to investigate the relationship between composition, viscosity and equilibration time-scales. These models have traditionally relied on the Fickian laws of diffusion with no explicit account of non-ideal interactions. We introduce the Maxwell-Stefan diffusion framework as an alternative method, which explicitly accounts for non-ideal interactions through activity coefficients. E-folding time is the time it takes for the difference in surface and bulk concentration to change by an exponential factor and was used to investigate the interplay between viscosity and solubility and the effect this has on equilibration time-scales within individual aerosol particles. The e-folding time was estimated after instantaneous increases in relative humidity to binary systems of water and an organic component. At low water mole fractions, viscous effects were found to dominate mixing. However, at high water mole fractions, equilibration times were more sensitive to a range in solubility, shown through the greater variation in e-folding times. This is the first time the Maxwell-Stefan framework has been applied to an atmospheric aerosol core-shell model and shows that there is a complex interplay between the viscous and solubility effects on aerosol composition that requires further investigation.

## 1 Introduction

Aerosol particles are an uncertain component of the Earth's atmosphere, interacting directly by scattering and absorbing radiation and indirectly by acting as nuclei for the formation of cloud droplets and ice crystals (Boucher et al., 2013). Properties of atmospheric aerosols depend upon their shape and composition, which can vary by orders of magnitude between individual particles. Predicting changes in the composition and micro-physics of individual aerosol particles is difficult, both theoretically and computationally, which is largely due to the complexity of chemical compositions and the need to account for multiple processes. Recent laboratory studies have found evidence that liquid-liquid phase separations can exist within aerosol particles (Zuend et al., 2010; Song et al., 2012). Further studies have found that organic aerosols can exist in an ultra-viscous or an amorphous state (Virtanen et al., 2010), where viscosities can range over many orders of magnitude (Lienhard et al., 2015). In an ultra-viscous state, mixing, or diffusion, through the particle is inhibited. There is conflicting evidence on the importance of the role that viscous aerosols play in the atmosphere according to focused laboratory studies on single particles and ensemble populations (Ye et al., 2016; Yli-Juuti et al., 2017). To better understand this process, a number of studies have modelled



individual aerosol particles to have a core-shell structure to model changing composition (Zobrist et al., 2011; Shiraiwa et al., 2013; O'Meara et al., 2016).

These core-shell models have relied upon Fickian laws of diffusion to simulate the mixing of compounds through individual particles. Fickian diffusion frameworks have played an important role in the investigation of mixing in glassy aerosol particles, where viscosity has been used as a proxy for phase state changes between a liquid and an ultra-viscous particle (Lienhard et al., 2015; Price et al., 2015). Diffusion in the Fickian sense is driven by a gradient in concentration and is described numerically by,

$$\frac{\partial c}{\partial t} = \nabla \cdot D \nabla c, \tag{1}$$

where $c(r,t)$ is the solute concentration, and $D$ the Fickian coefficient typically describing ideal diffusion properties (Fick, 1855). Equation 1 is Fick's second law and describes non-steady diffusion, which can vary with time. It has been shown that numerical models solving Fick's first and second laws for an aerosol particle give consistent solutions (O'Meara et al., 2016). Since the diffusion coefficient is an important factor in finding mixing time-scales, many studies have aimed to quantify this value, relying on inverting such models (Lienhard et al., 2015; Price et al., 2015, 2016).

Direct measurements of diffusion coefficients rely on single particle equilibration time-scales and find the Fickian diffusion coefficient in binary mixtures (Price et al., 2014; Lienhard et al., 2014). These experimental findings have given an insight into the physical processes underpinning diffusion through aerosol particles, however the limited database of these coefficients can not represent the range and complexity of multi-component systems found in the atmosphere. There is also great uncertainty in measurements of diffusion, the current best estimates of diffusion rates for water through amorphous $\alpha$-pinene at low water activities span over four orders of magnitude (Price et al., 2015). Investigations into the equilibration time of aerosol particles have largely been based on arbitrary limiting values or related to the solution viscosity through the Stokes-Einstein equation. However, use of the Stokes-Einstein equation can cause diffusion coefficients to deviate by three orders of magnitude in non-dilute solutions and up to five orders of magnitude in amorphous materials from direct experimental measurements (Power et al., 2013).

Mixing rules provide mutual diffusion coefficients in mixtures and describe the relationship between diffusion and concentration. Diffusion coefficients of pure substances act as limiting values in these functions at the extremes of concentration and are referred to as self diffusion coefficients. Many different mixing rules have been suggested, from a simple constant relationship (O'Meara et al., 2016), to the Darken relation (Darken, 1948), or the Vignes relation (Vignes, 1966) and a sigmoidal relationship (Lienhard et al., 2014) between diffusion coefficient and concentration. Different mixing rules in numerical models have been found to affect how aerosol composition varies with time (O'Meara et al., 2016).

The limited database of diffusion coefficients and viscosities has restricted the testing of Fickian diffusion models. Although the Fickian framework has been used to successfully model simple binary mixtures (Song et al., 2016), these simple binary systems can not truly represent the complexity of atmospheric aerosol particles. The molar based concentrations, $c$, used in Equation 1 are not convenient forms of thermodynamic activity variables (Taylor and Krishna, 1993), which relate to the non-ideal effects of mixing. Current Fickian based models do not explicitly separate the effects of solubility and drag that is treated





within a Maxwell-Stefan model. The Fickian model therefore limits any investigation into the relative effects of solubility and drag on overall diffusion rate. Furthermore, when multicomponent systems are considered the Fickian model is not generally applicable (Krishna and Wesselingh, 1997). For these reasons we suggest that the Maxwell-Stefan diffusion laws could offer an alternative framework in which the effects of phase state and solubility are explicitly combined.

The Maxwell-Stefan diffusion equation differs from the Fickian case as mixing is driven by a gradient in chemical potential. The Maxwell-Stefan equation is given by,

$$x_i \nabla \ln a_i = -\sum_{j \neq i}^{N} \frac{c_i \mathbf{J}_j - c_j \mathbf{J}_i}{c^2 Đ_{ij}}, \tag{2}$$

where $x_i$, $a_i$, $c_i$ and $\mathbf{J}_i$ are the mole fraction, activity coefficients, concentration (molar density) and flux of the $i^{\text{th}}$ component respectively, $c$ is the molar density of the mixture, $Đ_{ij}$ is the Maxwell-Stefan diffusion coefficient of component $i$ through

component $j$ and $N$ is the total number of components. By solving Equation 2 on a spherically symmetric grid, the effect of solubility on mixing through aerosol particles is explicitly accounted for through the inclusion of activity coefficients.

    The aim of this study is to investigate the effect of solubility on diffusion time-scales, by comparing both Fickian and Maxwell-Stefan model simulations. Binary mixtures of a representative organic compound and water are used to investigate the sensitivities of these models to both self diffusion coefficient and solubility at room temperature. Self diffusion coefficients

are investigated in the range of $1 \times 10^{-9} \mathrm{m}^2 \, \mathrm{s}^{-1}$ to $1 \times 10^{-25} \mathrm{m}^2 \, \mathrm{s}^{-1}$. To test the non-ideal effects of diffusion, sucrose was selected to represent a soluble compound and a series of monocarboxylic acids as examples of varying immiscibility in water.

## 2   Model description

The numerical diffusion frameworks are solved for the spherically symmetric shell model shown in Figure 1. The Fickian and Maxwell-Stefan laws of diffusion are solved to find the concentration fluxes on each of the shell boundaries. The diffusion

equations have been solved for multicomponent systems and details of these calculations can be found in Subsections 2.1 and 2.2. However, to investigate the sensitivities of diffusion time-scales to the model used, the number of components in the aerosol particle has been limited to two.

    The model has a moving boundary, which allows growth and shrinkage of the particle depending on the ambient conditions by assuming that equilibrium relative humidity equals the liquid water mole fraction in the outer aerosol shell. By assuming that

the outer aerosol shell is in equilibration with the ambient relative humidity, the study focuses solely on the effect non-ideality has on the rate of condensed phase diffusion and equilibration time-scales. To ensure that numerical effects did not accelerate the rate of diffusion, a maximum shell width is defined. When the radius of the particle grows or shrinks, the number of shells increases or decreases and the volume of the outer shell can vary depending upon the number of moles it contains. More details about the moving boundary can be found in the appendix.



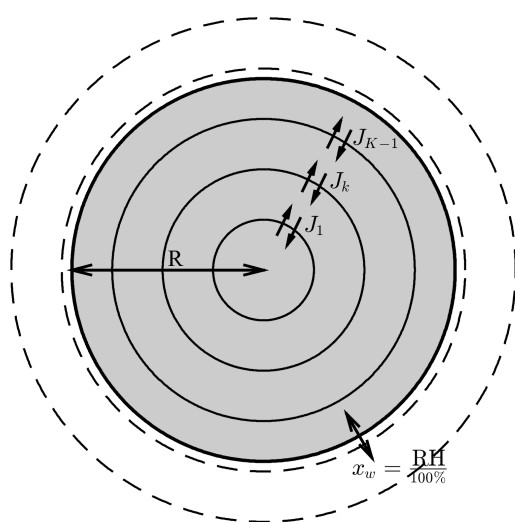

**Figure 1.** Shows the aerosol shell model, where diffusion equations are solved numerically to find concentration flux across the shell boundaries. The flux across the outer shell is set to zero, denoted as $J_K$. However, during each time step water is added to the outer shell to equilibriate it with the surrounding relative humidity, such that $x_w = \frac{\text{RH}}{100\%}$. For the purposes of this study it is assumed that the aerosol outer shell equilibrates instantaneously with the ambient conditions in order to investigate the effect of condense phase diffusion on equilibration time-scales. Notice that concentric shells have been initialised to limit the size that the shells grow to and the that the radius of the particle acts as a moving boundary for the outer shell.

## 2.1 Fickian framework

By assuming that concentration depends only on time and the radial component, Equation 1 was solved in a spherically symmetric coordinate system using the backward Euler method of finite differences. After rearranging, the change in concentration between the time steps is related by a tri-diagonal matrix of the form,

5      $$c_k^n = \alpha_k c_{k-1}^{n+1} + \beta_k c_k^{n+1} + \gamma_k c_{k+1}^{n+1},$$ (3)

where,

$$\alpha_k = -\frac{\Delta t \, r_{n-1}{}^2 D_{k-1}}{r_n{}^2 \Delta r_{n-\frac{1}{2}} \Delta r_n},$$

$$\beta_k = 1 - \frac{\Delta t \, r_{n-1}{}^2 D_{k-1}}{r_n{}^2 \Delta r_{n-\frac{1}{2}} \Delta r_n} + \frac{\Delta t \, r_{n+1}{}^2 D_{k+1}}{r_n{}^2 \Delta r_{n+\frac{1}{2}} \Delta r_n},$$

$$\gamma_k = -\frac{\Delta t \, r_{n+1}{}^2 D_{k+1}}{r_n{}^2 \Delta r_{n+\frac{1}{2}} \Delta r_n}.$$



The subscript $k$ corresponds to the shell number and superscript $n$ to the time step. Details of the steps taken during the matrix manipulation are given in the appendix. In the diffusion step, it was assumed that the organics were in-volatile, hence there is no external source of material diffusing through the drop, by specifying Neumann flux boundary conditions,

$$\frac{c_{K+1}^{n+1} - c_{K-1}^{n+1}}{\Delta r_K} = 0,$$
$$\frac{c_2^{n+1} - c_0^{n+1}}{\Delta r_1} = 0, \tag{4}$$

where the flux through the boundary of the $K^{\text{th}}$ shell is set to zero. Growth of the particle occurs when water is added to the surface shell, details of the moving boundary are given in the appendix.

## 2.2 Maxwell-Stefan framework

The Maxwell-Stefan law of diffusion from Equation 2 was solved on the aerosol shell model defined in Figure 1. During the diffusion step we assume that flux at the shell boundary is zero and consequently find a matrix to solve for the flux across each

of the shells given by,

$$\begin{bmatrix} x_1 \nabla \ln a_1 \\ x_2 \nabla \ln a_2 \\ \vdots \\ x_{N-1} \nabla \ln a_{N-1} \end{bmatrix} = \frac{A}{c^2} \begin{bmatrix} \mathbf{J}_1 \\ \mathbf{J}_2 \\ \vdots \\ \mathbf{J}_{N-1} \end{bmatrix}, \tag{5}$$

where A is a matrix defined as,

$$A_{ij} = \begin{cases} -\sum_{j \neq i}^{N} \left( \frac{c_j}{\DJ_{ij}} + \frac{c_i \rho_N M_i}{\DJ_{iN} \, \rho_i M_N} \right) & \text{if } i = j, \\ -\left( \frac{c_i}{\DJ_{ij}} - \frac{c_i \rho_N M_j}{\DJ_{iN} \, \rho_j M_N} \right) & \text{if } i \neq j. \end{cases} \tag{6}$$

Details of this calculation are given in the appendix. The solution matrix in Equation 5 is rearranged to find the diffusion fluxes,

$\mathbf{J}$, which are then substituted into Fick's first law,

$$\mathbf{J} = -D_k \frac{\partial \mathbf{c}}{\partial r_k}, \tag{7}$$

to find the corresponding diffusion coefficient, $D_k$, for $k^{\text{th}}$ shell boundary. The diffusion coefficientis then used in the numerical solution to Fick's second law in Equation 3.

    In our experiments the organic component is assumed to be involatile and only water enters and leaves the aerosol particle.

Water is added and removed from the particle at the end of each time step, which allows us to assume that the flux at the shell boundary is zero during the diffusion step. This study uses the UNIFAC group contribution model to estimate the activities, $a_i$, from Equation 2 (Fredenslund et al., 1975).



## 2.3 Diffusion coefficients

The Fickian and Maxwell-Stefan diffusion coefficients are both functions of concentration, temperature, pressure and composition. For higher order systems, the diffusion coefficient describing each component in a mixed solvent is required, this is mathematically expressed as a matrix of individual binary diffusion coefficients.

A mixing rule is used to estimate the relationship of a mutual diffusion coefficient with mole fraction. These mixing rules are based on self diffusion coefficients of pure substances, found at the limits of mole fraction, $D_{i,\text{self}}$. Diffusion coefficients in the literature cover a wide range of values (Shiraiwa et al., 2013; Price et al., 2015; Lienhard et al., 2015), hence the self diffusion coefficients in this study have been chosen to fall within this range in order to investigate model sensitivities.

     A variety of different mixing rules have been investigated (O'Meara et al., 2016), however for the purposes of this study, we
have selected the Darken and Vignes relations.

1. The Darken equation assumes a linear relationship between mole fraction and diffusion coefficient (Darken, 1948),

$$Ð_{ij} = x_i D_{i,\text{self}} + x_j D_{j,\text{self}}. \tag{8}$$

The Darken relation has been observed to better describe the mixing of ideal liquids (Wesselingh and Bollen, 1997).

2. The Vignes equation assumes a logarithmic relationship between mole fraction and diffusion coefficient (Vignes, 1966),

$$Ð_{ij} = \left(D_{i,\text{self}}\right)^{x_i} \left(D_{j,\text{self}}\right)^{x_j}. \tag{9}$$

The Vignes relation is preferred to describe the mixing of viscous fluids, where there is a large difference between their respective self diffusion coefficients (Wesselingh and Bollen, 1997).

The relationship between water mole fraction and the mutual diffusion coefficient found from the Darken and Vignes mixing
rules are shown in Figure 2.

     The Maxwell-Stefan framework separates the ideal and non-ideal effects of diffusion, unlike the Fickian model. Therefore, solutions to the Fickian and Maxwell-Stefan equations only coincide when the mixture is ideal and solubility does not affect the rate of mixing. The two frameworks are related through the so called thermodynamic factor, $\Gamma$,

$$D = Ð\Gamma. \tag{10}$$

The thermodynamic factor gives an effective Fickian diffusion coefficient for the Maxwell-Stefan case (Krishna and Wesselingh, 1997) and is a function of mole fraction and activity coefficient,

$$\Gamma_{ij} = \delta_{ij} + x_i \left. \frac{\partial \ln a_i}{\partial x_j} \right|_{T,p,x_k, k \neq j=1,2,\dots,n-1}, \tag{11}$$

evaluated at given conditions for temperature, $T$, pressure, $p$ and mole fraction, $x_i$, while keeping the mole fraction of all other species, $x_k$, constant (Taylor and Krishna, 1993), where $\delta_{ij}$ is the Kronecker delta. The resulting diffusion coefficient does





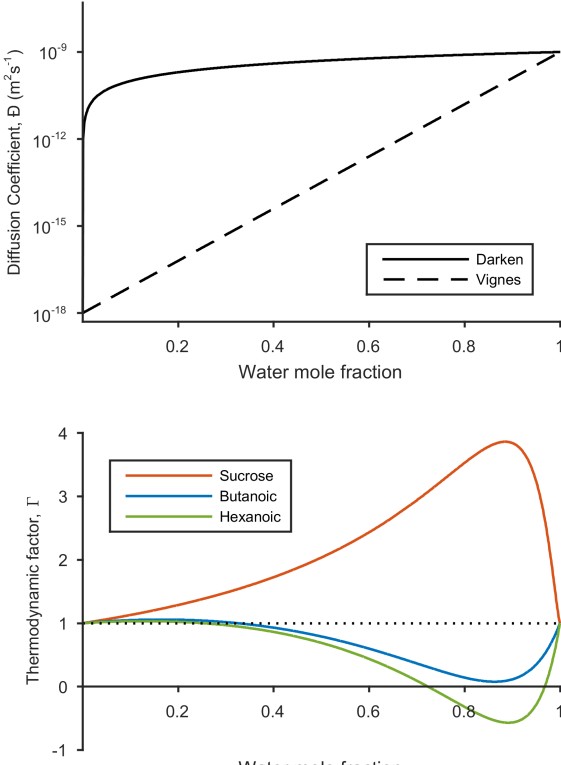

**Figure 2.** Shows the dependence of diffusion coefficient and thermodynamic factor on water mole fraction. The solid and dashed black lines represent the Darken and Vignes mixing rules from Equations 8 and 9 respectively. The self diffusion coefficients for water and the non-volatile species are $D_{\text{self}} = 1 \times 10^{-9}\,\text{m}^2\,\text{s}^{-1}$ and $D_{\text{self}} = 1 \times 10^{-18}\,\text{m}^2\,\text{s}^{-1}$ respectively. The thermodynamic factors for sucrose (red), butanoic acid (blue) and hexanoic acid (green) are shown in the lower panel. Phase transitions are indicated when the thermodynamic factor crosses the x-axis and equal zero. Activity coefficients used to calculate the thermodynamic factor were found using the UNIFAC model at a temperature of 293.15K.

provide a non-ideal correction to the Fickian diffusion coefficients, however it predicts negative diffusion coefficients when the thermodynamic factor becomes negative, which the Fickian framework has not been designed to deal with. The resulting diffusion coefficients are sensitive to the model used to calculate the activity coefficients (Taylor and Krishna, 1993), hence why we have not used the thermodynamic factor as a correction to Fick's laws. Instead, we show the relationship between the thermodynamic factor from Equation 10 and water mole fraction in Figure 2 to give an indication of how solubility effects differ between sucrose, butanoic and hexanoic acid.



## 3  Results

### 3.1  General model behaviour

In this study small self diffusion coefficients of the non-volatile organic compound have been used as a proxy for systems with a pronounced glassy state. To investigate the model sensitivities both the Darken and Vignes mixing rules were used with different values of the self diffusion coefficient to simulate different viscosities and various compounds were chosen to simulate the non-ideal effects of diffusion. We also investigated three different relative humidity ranges to consider how this affected the rate of mixing within individual aerosol particles.

Figures 3 and 4 show the general features of the diffusion frameworks through plots of changes in radial composition with time, after an instantaneous increase in equilibrium relative humidity at $t = 0$. In Figures 3 and 4 the self diffusion coefficient of the organic component has been kept constant at $1 \times 10^{-18} \mathrm{m}^2 \, \mathrm{s}^{-1}$, so as to focus on the effect of solubility on mixing time-scales and particle growth. The ideal Fickian solution acts as a control, where change in aerosol composition with time is modelled without solubility effects arising from intermolecular interactions. Then sucrose, butanoic and hexanoic acid provide examples with a spectrum of solubility in water to test sensitivity of diffusion rates to non-ideal effects.

By first considering the Fickian simulations in Figures 3 and 4 we are able to focus on the general model behaviour that relates to the ideal effects of diffusion, which in our simulations relate to the Darken and Vignes mixing rules respectively. There are two key differences between the simulations, the first is the timescale of diffusion and the second is the gradient of the diffusion front. These features have also been noted in previous modelling studies (O'Meara et al., 2016). At low water mole fractions, where the relative humidity is increased from 10% to 30% the Vignes simulations equilibrate on the scale of seconds, which is six orders of magnitude greater than the Darken mixing rule in Figure 3 predicts. However, at high water mole fractions, when the relative humidity is increased from 10% to 99% the timescales of diffusion are more alike. This difference in diffusion timescales between the two mixing rules can be related to the diffusion coefficients in Figure 2. At low water mole fractions the difference between the Darken and Vignes mixing rules is large, where as at high water mole fractions the difference between the two mixing rules is smaller.

Now we move on to discuss the Maxwell-Stefan simulations of sucrose, butanoic and hexanoic acids, which also take into account the non-ideal effects of diffusion through the activity coefficients from the UNIFAC model, which is assuming the liquid state. First notice that there is little difference between any of the simulations when relative humidity was instantaneously increased from 10% to 30% in the first column of both Figures 3 and 4. The effects of solubility have a negligible impact on the rate of diffusion at low water mole fractions, as Figure 2 shows that thermodynamic factors do not diverge significantly at water mole fractions less than 0.3.

In the cases where relative humidity has been instantaneously increased from 10% to 80%, the variation in growth rates and aerosol composition is greater than when relative humidity is increased to 30%. Variation in mixing time-scales can be explained by the diverging thermodynamic factors at high water mole fractions in Figure 2. It is the polarity of molecules in solution that determines their ability to mix and water molecules are particularly polar due to the position of hydrogen atoms and the permanent dipole moment. Polar molecules, such as sucrose are more likely to mix with water as intermolecular bonds



form between solute and solvent, which releases energy and further disrupts intermolecular forces between the solute particles and leads to mixing. In contrast, non-polar molecules or sections of molecules, such as alkyl chains, tend to aggregate within water due to hydrophobicity. Monocarboxylic acids are an example of this, as a COOH functional group is positioned at the end of an alkyl chain. As the length of the alkyl chain increases from butanoic to hexanoic we show that in both the Darken

and Vignes examples the rate of mixing reduces significantly due to the greater hydrophobic tendency.

As expected in Figures 3 and 4 we find that sucrose grows more quickly than the ideal Fickian case when relative humidity is increased from 10% to 80% or 99%, reaching equilibrium in almost half the time of the ideal case. The difference in equilibration times shows that solubility needs to be taken into consideration when modelling the change in composition of atmospheric aerosol particles with time, particularly at high water mole fractions.

Butanoic and hexanoic acid, which are the immiscible examples in Figures 3 and 4, are more interesting. The shorter chain monocarboxylic acid, butanoic acid, containing four carbon atoms, diffuses at a slower rate than the Fickian case as water on the surface of the particle is inhibited from diffusing into the centre of the particle. The slower rate of mixing corresponds to a thermodynamic factor which is between zero and one at high water mole fractions for butanoic acid shown in Figure 2. Figures 3 and 4 show that under an instantaneous increase in relative humidity from 10% to 80% and 99% an aerosol particle of

hexanoic acid does not grow and equilibrate with the ambient conditions. At high water mole fractions, Figure 2 shows that the thermodynamic factor for hexanoic acid is negative, corresponding to backwards diffusion against the concentration gradient, which is not expected in the Fickian sense of diffusion. We also find that butanoic acid equilibrates on a significantly slower time-scale when the Vignes mixing rule is used with the Maxwell-Stefan framework, which shows that there is a complex interplay between the viscous and soluble effects of diffusion that needs to be better understood.

The simulations in Figures 3 and 4 were chosen to demonstrate the efficacy of the numerical framework through their wide range of aqueous solubility. For such systems that potentially exhibit a range of amorphous states, it has been hypothesised that combination of absorption and adsorption is needed to explain observed hygroscopicity curves (Pajunoja et al., 2015). In each simulation here, diffusion is controlled by an equilibrated surface mole fraction and predictions from the UNIFAC model. For our model set up, there is a marginally larger water content in the hexanoic acid core after the 10-80% increase over the 10-99%

increase. For the Darken case in Figure 3, when relative humidity is increased to 80% the hexanoic particle equilibrates at a water mole fraction of 0.57, however when relative humidity is instantaneously increased to 99% the hexanoic core equilibrates to a water mole fraction of 0.54 and a shell of water mole fraction greater than 0.95 develops. We believe that the formation of a shell arises as the thermodynamic factor of hexanoic acid in Figure 2 equals zero at two points, at approximate water mole fractions of 0.7 and 0.95. When the thermodynamic factor vanishes at these two points no further diffusion takes place,

hence a clear discontinuity between the two different concentrations of the solution. This can not be referred to as a liquid-liquid phase separation as schlierens are not included in our simple model to initiate a second phase (Ciobanu et al., 2009). However, our findings still support laboratory evidence of liquid-liquid phase separations forming under conditions of high relative humidities (Renbaum-Wolff et al., 2016). In these studies the volatile liquid component is enclosed in the core of the aerosol particle, which suggests there are other surface effects that are not included into these simple diffusion models.

The slight difference in water mole fraction of the equilibrated hexanoic acid core is a result of our numerical approach to





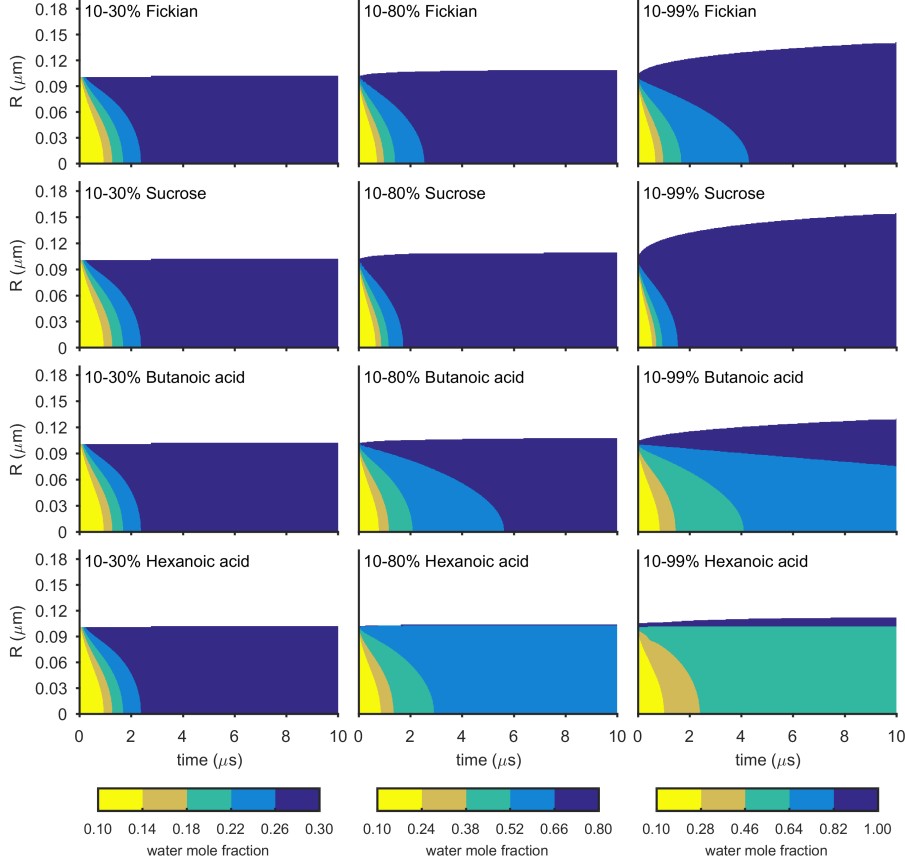

**Figure 3.** Shows the change in water mole fraction as a function of radius through time after an initial increase in relative humidity from 10% to 30%, 80% and 99%, at $t = 0$ in the left and right column respectively. Note that there is a different colour scale for each column to ensure clarity in the rate of mixing. The rows correspond to different non-volatile substances, with an arbitrary ideal species in the Fickian case, sucrose, butanoic acid and hexanoic acid from top to bottom. These simulations correspond to the Darken diffusion coefficient and thermodynamic factors given in Figure 2. The aerosol particles were initialised with a radius of $1 \times 10^{-7}$ m and the self diffusion coefficient of water and non-volatile as $D_{\text{self}} = 1 \times 10^{-9}$ m$^2$ s$^{-1}$ and $D_{\text{self}} = 1 \times 10^{-18}$ m$^2$ s$^{-1}$ respectively. The simulations were run at a temperature of 293.15K.

solving the diffusion equations for the aerosol framework. In order to find the flux across shell boundaries we use the average concentration of the shells on either side. This may also be the reason that the hexanoic cores equilibrate at lower water mole fractions than the predicted 0.7 based on the thermodynamic factor in Figure 2. In future work, we propose to combine the core numerical approach presented here with other processes likely taking place that need to to be considered with the history 5 of the particle and radial composition heterogeneity.





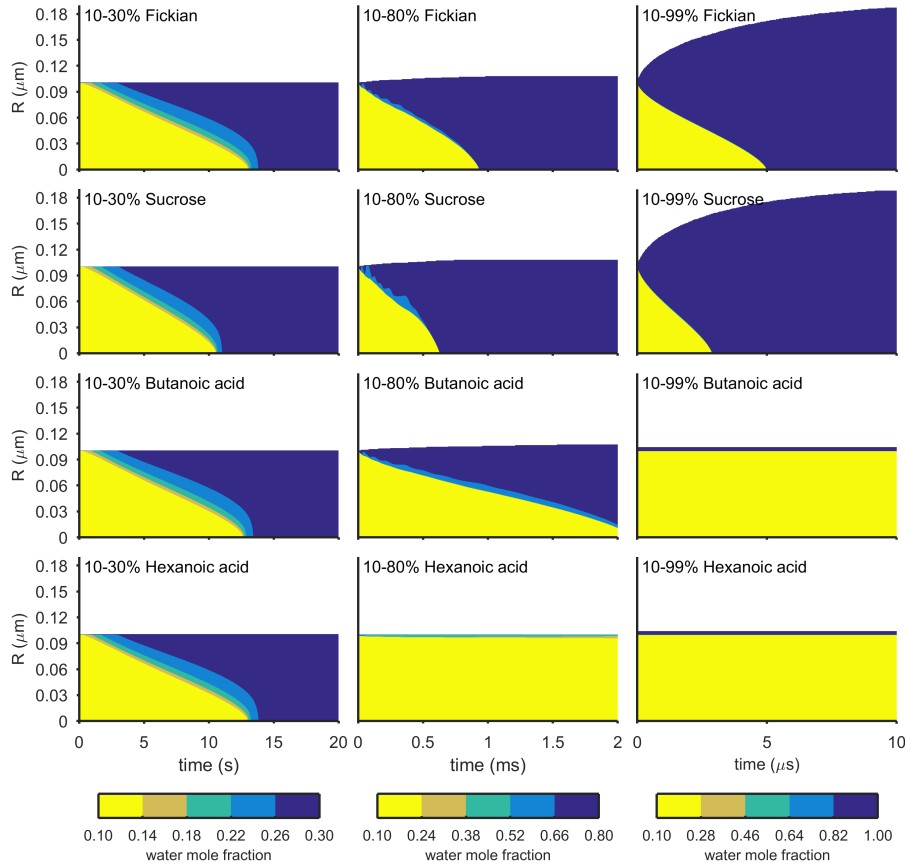

**Figure 4.** Shows the change in water mole fraction as a function of radius through time after an initial increase in relative humidity from 10% to 30%, 80% and 99%, at $t = 0$ in the left and right column respectively. Note that there is a different colour scale for each column to ensure clarity in the rate of mixing. The rows correspond to different non-volatile substances, with an arbitrary ideal species in the Fickian case, sucrose, butanoic acid and hexanoic acid from top to bottom. These simulations correspond to the Vignes diffusion coefficient and thermodynamic factors given in Figure 2. The aerosol particles were initialised with a radius of $1 \times 10^{-7}$ m and the self diffusion coefficient of water and non-volatile as $D_{\mathrm{self}} = 1 \times 10^{-9}\,\mathrm{m}^2\,\mathrm{s}^{-1}$ and $D_{\mathrm{self}} = 1 \times 10^{-18}\,\mathrm{m}^2\,\mathrm{s}^{-1}$ respectively. The simulations were run at a temperature of 293.15K.

### 3.2 Model sensitivities

E-folding time has been used previously by Zaveri et al. (2014) and O'Meara et al. (2016) to comparing equilibration time-scales. The e-folding time is defined as the time it takes for the difference in surface and bulk concentration to change by an exponential factor. Details of this metric and how it is calculated can be found in the appendix. Figure 5 shows the results of the sensitivity of e-folding time to viscosity and solubility. Solubility has been varied through activity coefficients in Equation 2 and the viscosity of the solution through self diffusion coefficients in Equations 8 and 9. For this study the self diffusion



coefficient of water has been kept constant at $2 \times 10^{-9}\,\mathrm{m^2\,s^{-1}}$ and the self diffusion coefficient of the second compound has been varied along the x-axis. The y-axis shows the effect of initial particle radius on e-folding time. Figures 3 and 4 showed that hexanoic acid did not equilibrate when relative humidity is increased instantaneously from 10% to 80% and therefore does not reach the e-folding time criteria. Hence sucrose and butanoic acid have been used to show how a spectrum of solubility can

effect equilibration times through the red and blue lines respectively, in Figure 5. Throughout the experiment the equilibration time-scales were compared back to the Fickian solution, shown by the black dashed line in Figure 5, which does not take into account the non-ideal effects of diffusion.

Figure 5 shows e-folding time contours as a function of aerosol self diffusion coefficient and initial radius. We found that by changing the mixing rule between Darken and Vignes affected the gradient of the e-folding line contour. In the Darken

case, e-folding contours are horizontal indicating that equilibration times are independent of the self diffusion coefficient of the non-volatile organic compound and the size of the particle has the biggest influence on equilibration time. By referring back to Figure 2 we see that the Darken mutual diffusion coefficient on a logarithmic scale of diffusion coefficients is around $1 \times 10^{-9}\,\mathrm{m^2\,s^{-1}}$, which is the self diffusion coefficient of water, the volatile component. In comparison, the Vignes mixing rule gives e-folding time contours with a positive gradient in Figure 5, as a result of the more significant role of the self diffusion

coefficient of the non-volatile component in Equation 9.

Through investigating both high and low water mole fractions by increasing the relative humidity from 10% to 30% and 80% in Figure 5 we show how the spectrum of solubility translates to equilibration time-scales. In both the Vignes and Darken cases solubility does not show a spread of equilibration times when water mole fractions are low and relative humidity is increased from 10% to 30%. However, when relative humidity is increased from 10% to 80% solubility becomes a more important factor

as the grey area shows the spread in the conditions for the e-folding time contours.

Figure 5 also highlights the plasticising effect of water within the aerosol system, as at low water mole fractions equilibration times are significantly longer than at high water mole fractions, especially in the Vignes case. Here we have shown that there is a complex interaction between viscosity, solubility and humidity of equilibration time-scales, which is important to understand due to the abundance on water in the atmosphere.

**4 Conclusions**

The main aim of this study has been to introduce the Maxwell-Stefan law of diffusion to describe the changing composition of atmospheric aerosol particles with time. The Maxwell-Stefan equation could act as an alternative framework to the widely used Fickian framework, which has limitations as it does not inherently account for solubility effects. From comparing the sensitivities of these models we found:

– Observed aerosol partitioning in laboratory studies can not be replicated using a Fickian framework, which is driven by a gradient in concentration without modifying the Fickian diffusion coefficient to account for the non-ideal effects.





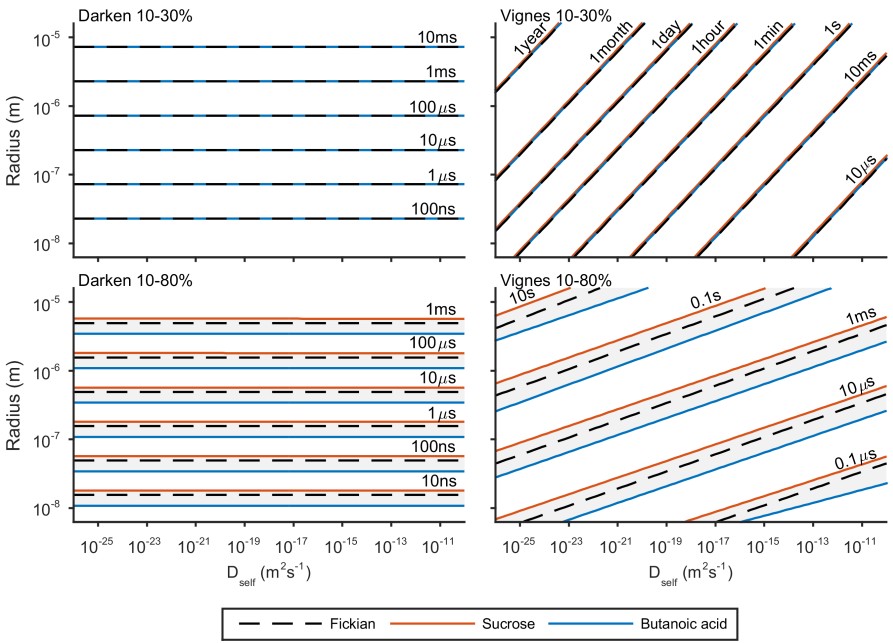

**Figure 5.** Shows e-fold time contours for both the Darken and Vignes mixing rules and how they depend on initial particle radius and self diffusion coefficient of the non-volatile organic compound. In both cases e-folding times have been found for increases in relative humidity from 10% to 30% and 80%. The black dashed line represents the e-fold contour found using the Fickian framework. The grey shaded area represents the sensitivity in the model due to the range in solubility between sucrose and butanoic acid given by the red and blue lines respectively. The simulations were run at a temperature of 293.15K with initially 30 shells equally distributed across the radius of the particle.

- Including solubility effects arising from intermolecular interactions, are essential to model sustained component separations within aerosol particles. The Maxwell-Stefan framework accounts for these through activity coefficients, calculated using the UNIFAC model.

- At low water mole fractions, viscosity was shown to be the most influential factor on equilibration times within aerosol particles.

- At high water mole fractions, the variation in equilibration time-scales is due to solubility effects, which is especially significant in the atmosphere where there is an abundance of water.

- Through simple binary systems of water and a non-volatile secondary organic aerosol, we have shown that there is a complicated relationship between the viscous and soluble effects of mixing. Atmospheric particles are far more complicated systems, with a far greater number of components, all with differing properties. Therefore it is essential to ensure the most suitable framework is used to model this system.



This area of research aims to understand the key micro-physical processes that underpin cloud development and therefore produce models that better predict these processes. We found that at low water mole fractions, equilibration times were most sensitive to changes in viscosity, however at high water mole fractions solubility became a more important factor to consider. This could have significant implications for atmospheric processes. Especially, the activation of cloud condensation nuclei and
ice nuclei, which occur at high water mole fractions.

This study highlights one key question that needs to be addressed before we continue to investigate the impact of partitioning within aerosol particles on atmospheric models, which is whether current frameworks used to model aerosol composition are suitable to apply to highly complex atmospheric systems. The Fickian model works well for simple two component systems, where diffusion coefficients can be measured directly. However, for complex systems with multiple components, the Maxwell-
Stefan framework offers an alternative which allows mixing against the concentration gradient and phase separations to form.

The Fickian approach has been preferred, as direct measurements of mixing based on equilibration time-scales gives a mutual Fickian diffusion coefficient. However, the application of these binary diffusion coefficients to complex, multicomponent atmospheric systems is questionable. On the other hand, the Maxwell-Stefan laws are inherently multicomponent. The difficulty we have is selecting the most appropriate mixing rule, which relates a mutual diffusion coefficient to mole fraction as
a function of the self diffusion coefficients at infinite dilution. Predictive models for Maxwell-Stefan diffusivities have been found experimentally from self diffusion coefficients (Xin, 2013) and can also be accessed theoretically using molecular dynamic simulations in the region of infinite dilution, where thermodynamic factors can be neglected (Xin, 2013; Krishna and van Baten, 2010a, b). In this study the Darken and Vignes mixing rules have been investigated, but alternatives such as a constant relationship or a sigmoidal relationship should also be considered (O'Meara et al., 2016). Further investigation is needed to
decide which of the relationships are most appropriate and also if they capture the complexity of multi-component systems under a range of conditions.

To ultimately answer the question, whether current frameworks describing aerosol composition successfully model atmospheric processes, more laboratory studies are required to test model predictions. Through these investigations, we will then be able to better understand the competition between viscosity (or phase state) and solubility to dominate partitioning within
atmospheric aerosol particles.

*Code availability.*  TEXT

*Data availability.*  TEXT

*Code and data availability.*  TEXT





## Appendix A: Numerical Method

### A1 Solving Fick's second law of diffusion using the backward Euler method

Fick's second law predicts how concentration changes with time,

$$\frac{\partial c}{\partial t} = \nabla \cdot D \nabla c, \tag{A1}$$

where $c(r,t)$ is the solute concentration. For an aerosol particle we assume that the concentration only depends upon the radius in a spherically symmetric coordinate system,

$$\frac{\partial c}{\partial t} = \frac{1}{r^2}\frac{\partial}{\partial r}\left(Dr^2\frac{\partial c}{\partial r}\right). \tag{A2}$$

In order to solve this numerically the backward Euler method of finite differences was applied to give,

$$\frac{c_k^{n+1} - c_k^n}{\Delta t} = \frac{1}{r_k^2 \Delta r_k}\left[D_{k+1}r_{k+1}^2\left(\frac{c_{k+1}^{n+1} - c_k^{n+1}}{\Delta r_{k+\frac{1}{2}}}\right) - D_{k-1}r_{k-1}^2\left(\frac{c_k^{n+1} - c_{k-1}^{n+1}}{\Delta r_{k-\frac{1}{2}}}\right)\right]. \tag{A3}$$

Rearranging this equation so all future time steps appear on one side of the equation gives a tri-diagonal matrix of the form,

$$c_k^n = \alpha_k c_{k-1}^{n+1} + \beta_k c_k^{n+1} + \gamma_k c_{k+1}^{n+1}, \tag{A4}$$

where,

$$\alpha_k = -\frac{\Delta t\, r_{k-1}{}^2 D_{k-1}}{r_k{}^2 \Delta r_{k-\frac{1}{2}} \Delta r_k},$$

$$\beta_k = 1 - \frac{\Delta t\, r_{k-1}{}^2 D_{k-1}}{r_k{}^2 \Delta r_{k-\frac{1}{2}} \Delta r_k} + \frac{\Delta t\, r_{k+1}{}^2 D_{k+1}}{r_k{}^2 \Delta r_{k+\frac{1}{2}} \Delta r_k},$$

$$\gamma_i = -\frac{\Delta t\, r_{k+1}{}^2 D_{k+1}}{r_k{}^2 \Delta r_{k+\frac{1}{2}} \Delta r_k}.$$

To solve this equation it was assumed that there was no external source of material diffusing through the drop. By specifying

Neumann boundary conditions, where the flux through the shell boundary is set to zero. The flux conditions are given as,

$$\frac{c_{K+1}^{n+1} - c_{K-1}^{n+1}}{\Delta r_K} = 0,$$

$$\frac{c_2^{n+1} - c_0^{n+1}}{\Delta r_1} = 0, \tag{A5}$$

which means that elements in the first and last rows of the matrix are adjusted to agree with this condition.

### A2 Solving Maxwell-Stefan equation for an aerosol particle

We begin with the Maxwell-Stefan equation,

$$x_i \nabla \ln a_i = -\sum_{j=1, j\neq i}^{N} \frac{c_i \mathbf{J}_j - c_j \mathbf{J}_i}{c^2 \DJ_{ij}}. \tag{A6}$$



By specifying that the $N^{\text{th}}$ volume flux is equal to the negative sum of all other volume fluxes, therefore not allowing an external source of material to enter the particle we can define a reference frame in which to solve the problem. Rewriting the equation, this gives,

$$x_i \nabla \ln a_i = \frac{c_i \mathbf{J}_N}{c^2 Đ_{iN}} - \sum_{j=1, j \neq i}^{N} \frac{c_j \mathbf{J}_i}{c^2 Đ_{ij}} + \sum_{j=1, j \neq i}^{N-1} \frac{c_i \mathbf{J}_j}{c^2 Đ_{ij}}, \tag{A7}$$

5   where,

$$\mathbf{J_N} = -\frac{\rho_N}{M_N} \sum_{j=1}^{N-1} \frac{M_j \mathbf{J}_j}{\rho_j},$$

$$= -\frac{\rho_N}{M_N} \frac{M_i \mathbf{J}_i}{\rho_i} - \frac{\rho_N}{M_N} \sum_{j \neq 1}^{N-1} \frac{M_j \mathbf{J}_j}{\rho_j}, \tag{A8}$$

where $\rho_i$ and $M_i$ correspond to the density and mass of shell $i$. After substituting this into Equation (A7) yields,

$$x_i \nabla \ln a_i = \frac{\mathbf{J}_i}{c^2} \left( \frac{c_i \rho_N M_i}{Đ_{iN} \, \rho_i M_N} - \sum_{j=1, j \neq i}^{N} \frac{c_j}{Đ_{ij}} \right) + \sum_{j=1, j \neq i}^{N-1} \frac{\mathbf{J_j}}{c^2} \left( \frac{c_i}{Đ_{ij}} - \frac{c_i \rho_N M_j}{Đ_{iN} \, \rho_j M_N} \right). \tag{A9}$$

The numerical model is solved using matrix algebra and written in that form gives,

$$\begin{bmatrix} x_1 \nabla \ln a_1 \\ x_2 \nabla \ln a_2 \\ \vdots \\ x_{N-1} \nabla \ln a_{N-1} \end{bmatrix} = \frac{\mathrm{A}}{c^2} \begin{bmatrix} \mathbf{J}_1 \\ \mathbf{J}_2 \\ \vdots \\ \mathbf{J}_{N-1} \end{bmatrix}, \tag{A10}$$

where A is a matrix defined as,

$$A_{ij} = \begin{cases} -\sum_{j \neq i}^{N} \left( \frac{c_j}{Đ_{ij}} + \frac{c_i \rho_N M_i}{Đ_{iN} \, \rho_i M_N} \right) & \text{if } i = j, \\[2ex] -\left( \frac{c_i}{Đ_{ij}} - \frac{c_i \rho_N M_j}{Đ_{iN} \, \rho_j M_N} \right) & \text{if } i \neq j. \end{cases} \tag{A11}$$

## A3   Moving boundary

15   Each time step is separated into a diffusion step and a moving boundary step, which allows the amount of water in the particle to change. This was not used during the model runs in this paper, however the development of a moving boundary is vital for the inclusion of a diffusion step into a parcel model and to investigate the effect of aerosol composition on cloud micro-physical properties.

If the change in volume is positive and water is deposited onto the surface of the aerosol particle then the steps involved to
20   change the outer boundary of the particle are as follows:





1. The new radius, $R$, of the particle is found, based on the change in volume, $\Delta V$.

2. The change in volume is then used to calculate the molar concentration in the outer shell as follows:

$$x_w^{n+1} = x_w^n + \frac{\Delta V \rho_w}{M_w}, \tag{A12}$$

   where $n$ corresponds to the time step.

3. Shell boundaries are then initiated and filled with the total moles in each layer. If the final shell is filled the molar concentration is distributed over to a new shell, see Figure 1.

4. Shell boundaries are initially fixed distances apart, however in the final step, the outer shell radius, $R_K$ is changed to the same as the new radius, $R$, calculated in the first step, such that $R_K = R$, as in Figure 1.

If the change in volume is negative and water is being removed from the particle surface, then the steps involved to change the outer boundary of the particle are as follows:

1. The total volume of water is calculated as sum of the volume of each individual shell $V_k$, the molar concentration of each shell $c_k$, molar mass of water $M_w$ and the density of water $\rho_w$,

$$V_w = \sum_{k=1}^{K} \frac{c_k V_k M_w}{\rho_w}. \tag{A13}$$

2. Only water is removed from the aerosol particle, therefore if $V_w = 0$ then there are no changes to the particle volume, if $V_w < \Delta V$ then all of the water is removed from the particle and if $V_w > \Delta V$, water is first removed from the outer shells and replaced by with other components.

3. After the water has been removed, a new outer shell radius is found,

$$R_K = \left( R_{K-1}{}^3 - \frac{3 M_s N_s}{4 \pi \rho_s} \right)^{\frac{1}{3}}, \tag{A14}$$

   where $R_K$ is the new radius of the outer shell and the subscript corresponds the number of the shell, $N_s$ is the number of moles of solute in the $K^{\text{th}}$ shell, $M_s$ and $\rho_s$ are the molar mass and density of the compound respectively, see Figure 1.

**Appendix B: Model testing**

**B1   Model initialisation**

This study has assumed that the mole fraction of water in the condensed phase in the outer shell is in equilibration with the ambient saturation relative humidity. This assumes ideality of the accommodation coefficient, which enables the study to focus on the sensitivity of diffusion time-scales to the framework used. The equation for water mole fraction,

$$x_w = \frac{N_w}{N_w + N_s}, \tag{B1}$$





where $N_w$ and $N_s$ are the number of molecules of water and the solute in the outer shell respectively, and the equation for the volume of the outer shell,

$$V = N_w \left( \frac{M_w}{\rho_w} \right) + N_s \left( \frac{M_s}{\rho_s} \right) \tag{B2}$$

were used to find the volume of water to be added or removed from the system in order to keep the outer shell equilibrated with the saturation relative humidity. In this study the molar mass and density of the solute were kept constant at $M_s = 400 \text{g mol}^{-1}$ and $\rho_s = 1.5 \times 10^3 \text{kg m}^{-3}$. For water these values were $M_w = 18 \text{g mol}^{-1}$ and $\rho_s = 1 \times 10^3 \text{kg m}^{-3}$.

**B2   E-folding time**

To find the e-folding time of a system, first the ratio $Q$ is defined as,

$$Q = \frac{\left| C_s - \overline{C}_b \right|_{t \geq 0}}{\left| C_s - \overline{C}_b \right|_{t=0}}, \tag{B3}$$

where $C_s$ is the surface shell concentration and $\overline{C}_b$ bulk concentration (O'Meara et al., 2016). The e-folding time is then defined as the time when the difference between surface shell and bulk concentration changes by an exponential factor. Numerically this is $Q = e^{-1}$.

*Acknowledgements.* This work was funded by the Natural Environment Research Council (NERC) through the PhD studentship of Kathryn Fowler under the grant reference number NE/L002469/1.





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
