# Peer review of "Maxwell-Stefan diffusion: a framework for predicting condensed phase diffusion and phase separation in atmospheric aerosol"

_Atmospheric Chemistry and Physics, 2017_

## Referee Comment (RC1) · Anonymous Referee #1 · 15 Sep 2017

This study presents the Maxwell-Stefan diffusion framework that explicitly treats non-ideal interactions of components by the use of activity coefficients. by relaxing the Fickian law of diffusion. The authors demonstrate a complex interplay between the viscous and solubility effects on chemical composition. Currently it is urgent to resolve effects of phase state and phase separation on various gas-particle interactions and this paper represents an important contribution to the community. The manuscript is well written and their methods and results seem reasonable. I support publication of this manuscript in ACP after below a few points are addressed.

- Introduction is written very well. Just one point: it is claimed that there is currently

[Figure]

no models that treat solubility and diffusion explicitly and separately. However, kinetic multi-layer models based on the PRA framework (Poschl et al., 2007) such as KM-GAP (Shiraiwa et al., 2012) do treat diffusion and solubility by constraining transport from surface to bulk using solubility, and also considers non-ideality by using activity coefficients (Shiraiwa et al., 2013). This could be acknowledged.

- The model assumes that the particle outer shell equilibrates instantaneously with ambient conditions. How good is this assumption? I suspect this assumption might invalid for glassy particles under low temperatures when water diffusivity drops substantially. This is also related to adsorptive vs. absorptive water uptake. I appreciate if you extend discussions on this issue.

- This study presents simulations results for binary system of water and one organic component. I wonder if this method also works well for complex multi-component mixtures as expected in ambient organic particles. I guess it should work in theory, but would you expect any practical challenges such as difficulty on finding converged solutions or computational time?

---

## Referee Comment (RC2) · Anonymous Referee #2 · 2 Nov 2017

Review of "Maxwell-Stefan diffusion: a framework for predicting condensed phase diffusion and phase separation in atmospheric aerosol" by Fowler et al.

This article proposes a Maxwell-Stefan diffusion framework in an atmospheric core-shell model for the interactions of compositions of aerosols with water vapors. This is an interesting and important topic because aerosol microphysical process affects many processes in the atmosphere, such as atmospheric cloud formations and eventually the global energy budget. And more importantly, there are still large uncertainties in our current climate models to represent this. It is a challenge for climate models to appropriately represent the aerosol formations and activation. This study is well within the scope of ACP, with sound model design. It deserves the publication in ACP after all of my comments are properly addressed.

Page 1, line 17: the size or mass of aerosols vary by orders of magnitude? This is need to be clearly stated.

Page 2, line 9: "D the Fickian coefficient typically describing ideal diffusion properties' should be changed to "D is the Fickian coefficient that describes the ideal diffusion properties".

Page 2, line 11: what is Fick's first and second laws? Equations (1) and (3) respectively? Readers need a clear message here.

Page 2, line 16: there should be a "," after however.

Page 2, lines 17-19: "There is also great uncertainty in measurements of diffusion, the current best estimates of diffusion rates……" should be changed to "There is also great uncertainty in measurements of diffusion. For instance, the current best estimates of diffusion rates……"

Page 2, lines 20: I suggest that the authors should explicitly describe the Stokes-Einstein equation here because most of readers are not familiar with this.

Page 3, line 2: "Furthermore, when multicomponent systems are considered the Fickian model is not generally……" should be changed to "Furthermore, when multicomponent systems are considered, the Fickian model is not generally……".

Page 3, line3: add a comma, after "For these reasons".

Page 4, Figure 1: the caption of last sentence seems to be confusing. Please re-write.

Page 5, line 17: change the typos of "coefficientis" to "coefficient is".

Page 5, line 21: what is UNIFAC? This needs to be briefly explained.

Page 5, line 21: change "activities" to "activity coefficient".

Page 6, line 9: add comma after "however".

Page 7, caption of Figure 2: change "equal" as "equals".

Page 8, line 4: add a comma after "To investigate the model sensitivities".

Page 8, line 9 and line 14: again, add a comma after "Figures 3 and 4".

Page 8, lines 5-6: what are the different values of the self diffusion coefficient? what are the three different relative humidity? Authors need to explain them clearly here.

Page 12, line 9: "by" should be deleted here.

Page 12, line 10: there should be a comma between "horizontal" and "indicating".

Page 12, line 12 and 17: again, comma needs to be added after "Figure 2" in line 12 and "Figure 15 in line 17.

Page 19, lines 30-31: O'Meara et al. (2016) should be updated from ACPD to ACP.

---

## Author Comment (AC1) · 10 Nov 2017

Please find attached the authors' response to comments received by anonymous referees 1 and 2.

Please also note the supplement to this comment: https://www.atmos-chem-phys-discuss.net/acp-2017-424/acp-2017-424-AC1-supplement.pdf

---

## Author Response (AR1)

**Response to Reviewer Comments**

Kathryn Fowler[1], Paul J. Connolly[1], David O. Topping[1], and Simon O'Meara[1]

[1]University of Manchester, School of Earth, Atmospheric and Environmental Science

*Correspondence to:* Paul James Connolly (paul.connolly@manchester.ac.uk)

**We would like to thank the referees for their positive and constructive comments, which we have used to improve our manuscript. Attached is a PDF showing where changes have been made to the original version of the submitted manuscript and below are our specific responses to the referee comments.**

**Response to: Anonymous Referee #1**

Introduction is written very well. Just one point: it is claimed that there is currently no models that treat solubility and diffusion explicitly and separately. However, kinetic multi-layer models based on the PRA framework (Pöschl et al., 2007) such as KM-GAP (Shiraiwa et al., 2012) do treat diffusion and solubility by constraining transport from surface to bulk using solubility, and also considers non-ideality by using activity coefficients (Shiraiwa et al., 2013). This could be acknowledged.

We have acknowledged that the KM-GAP model includes the non-ideal effects of diffusion on page 3 lines 9-11.

The model assumes that the particle outer shell equilibrates instantaneously with ambient conditions. How good is this assumption? I suspect this assumption might invalid for glassy particles under low temperatures when water diffusivity drops substantially. This is also related to adsorptive vs. absorptive water uptake. I appreciate if you extend discussions on this issue.

We have extended our discussion and referred to Mikhailov et al. (2009) on page 4 lines 4-8.

This study presents simulations results for binary system of water and one organic component. I wonder if this method also works well for complex multi-component mixtures as expected in ambient organic particles. I guess it should work in theory, but would you expect any practical challenges such as difficulty on finding converged solutions or computational time?

We have developed the framework to deal with complex multi-component systems, however in this paper have decided to use simple binary systems here to demonstrate the key sensitivities of the Maxwell-Stefan framework. The Maxwell-Stefan framework is inherently multi-component in the form we have used in Equation 2 from the paper. Running the model for more than two components does not cause any theoretical problems, however computational time increases by a small amount.

**Response to: Anonymous Referee #2**

Page 1, line 17: the size or mass of aerosols vary by orders of magnitude? This is need to be clearly stated.

We have clarified that the size can vary by orders of magnitude on line 17.

Page 2, line 9: "D the Fickian coefficient typically describing ideal diffusion properties" should be changed to "D is the Fickian coefficient that describes the ideal diffusion properties".

Agreed.

Page 2, line 11: what is Fick's first and second laws? Equations (1) and (3) respectively? Readers need a clear message here.

We have restructured the paragraph to emphasise that current models that use Fick's first and second laws give consistent solutions, introducing both of the Fickian laws here on page 2 line 7. We have then further clarified that the Maxwell-Stefan framework that the first law was used convert flux, **J** to a Fickian diffusion coefficient on page 6 lines 6-7.

Page 2, line 16: there should be a "," after however.

Included.

Page 2, lines 17-19: "There is also great uncertainty in measurements of diffusion, the current best estimates of diffusion rates......" should be changed to "There is also great uncertainty in measurements of diffusion. For instance, the current best estimates of diffusion rates......"

Agreed.

Page 2, lines 20: I suggest that the authors should explicitly describe the Stokes-Einstein equation here because most of readers are not familiar with this.

More detail of the Stokes-Einstein equation has been included on page 2, line 10, however we feel that including the equation explicitly is unnecessary to communicate that in many cases diffusion coefficients have not been directly measured causing a great variation in the magnitude of diffusion coefficients in the current literature.

Page 3, line 2: "Furthermore, when multicomponent systems are considered the Fickian model is not generally......" should be changed to "Furthermore, when multicomponent systems are considered, the Fickian model is not generally......".

Second comma has been included.

Page 3, line3: add a comma, after "For these reasons".

Agreed.

Page 4, Figure 1: the caption of last sentence seems to be confusing. Please re-write.

The sentence has been changed to provide more clarity.

Page 5, line 17: change the typos of "coefficientis" to "coefficient is".

Corrected.

Page 5, line 21: what is UNIFAC? This needs to be briefly explained.

More explanation has been included in lines ... .

Page 5, line 21: change "activities" to "activity coefficient".

Amended.

Page 6, line 9: add comma after "however".

Included.

Page 7, caption of Figure 2: change "equal" as "equals".

Amended.

Page 8, line 4: add a comma after "To investigate the model sensitivities".

Included.

Page 8, line 9 and line 14: again, add a comma after "Figures 3 and 4".

Included.

Page 8, lines 5-6: what are the different values of the self diffusion coefficient? what are the three different relative humidity? Authors need to explain them clearly here.

Agreed. More clarity has now been included in lines ... .

Page 12, line 9: "by" should be deleted here.

Amended.

Page 12, line 10: there should be a comma between "horizontal" and "indicating".

Included.

Page 12, line 12 and 17: again, comma needs to be added after "Figure 2" in line 12 and "Figure 15 in line 17.

Included.

Page 19, lines 30-31: O'Meara et al. (2016) should be updated from ACPD to ACP.

Citation updated. See references for O'Meara et al. (2016).

Shiraiwa, M., Pfrang, C., Koop, T., and Pöschl, U.: Kinetic multi-layer model of gas-particle interactions in aerosols and clouds (KM-GAP): linking condensation, evaporation and chemical reactions of organics, oxidants and water, 
[revised manuscript text omitted]